# Evaluation of a Multi-Isotope Approach as a Complement to Concentration Data within Environmental Forensics

Simon Pontér [1,2,*], Stacy Sutliff-Johansson [1], Emma Engström [1,2], Anders Widerlund [1], Anna Mäki [3], Katerina Rodushkina [2,†], Cora Paulukat [2] and Ilia Rodushkin [1,2]

1 Division of Geosciences and Environmental Engineering, Luleå University of Technology, S-971 87 Luleå, Sweden; stacy.sutliff-johansson@ltu.se (S.S.-J.); emma.engstrom@ltu.se (E.E.); anders.widerlund@ltu.se (A.W.); Ilia.Rodushkin@alsglobal.com (I.R.)
2 ALS Laboratory Group, ALS Scandinavia AB, Aurorum 10, S-977 75 Luleå, Sweden; katerina.rodiouchkina@ugent.be (K.R.); cora.paulukat@alsglobal.com (C.P.)
3 Vatten & Miljökonsulterna AB, Aurorum 2, S-977 75 Luleå, Sweden; anna.maki@vmkonsulterna.se
* Correspondence: simon.ponter@ltu.se
† Present address: Atomic and Mass Spectrometry Research Group, Department of Chemistry, Ghent University, Campus Sterre, Krijgslaan 281-S12, B-9000 Ghent, Belgium.

**Abstract:** Heavy metal contamination was identified in groundwater monitoring wells surrounding a waste deposit facility at the Rönnskär Cu–Pb–Zn smelter in Skellefteå, Northern Sweden, as well as in brackish water and sediments from the nearby harbor. Following an investigative study of the surrounding area, brackish water from the Baltic Sea and sediments from a nearby harbor were also determined to be contaminated. This study investigated the ranges of isotopic compositions of four elements (Cd, Cu, Pb, and Zn) in smelter materials (ores, products, and waste) and polluted groundwater sediments of the affected area. The study's objective was to evaluate the variability of the polluting source and identify possible isotope fractionation. This study further assesses the viability of using isotopic information to identify the source of the pollutant. These data were used in combination with multi-element screening analysis and multivariate statistical techniques. Expanding the number of elements utilized in isotope tracing empowers our abilities to decipher the source(s) and the extent of environmental exposure from contamination events related to mining and refining operations.

**Keywords:** isotope ratios; smelter; natural variability; fractionation groundwater contamination; heavy metals



## 1. Introduction

Environmental Forensics is a scientific methodology developed for identifying sources, the timing of release, and transport pathways for potentially hazardous environmental contaminants. It combines a variety of analytical methods with principles derived from disciplines such as chemistry, geology, geochemistry, hydrogeology, and statistics, with the purpose to provide objective scientific and legal conclusions on the source and/or time of a contaminant release into the environment [1–3]. The field has gained significant attention in the last few decades as the need to identify contaminant sources and dispersal pathways in our environment continues to grow. Such understanding is essential to mitigate the potentially harmful effects on ecosystems and on human health as well as to optimize remediation strategies [1–5].

Industrial activities such as mining and refining operations discharge a wide range of contaminants, but the most significant metalloids and trace metals are As, Cu, Cd, Hg, Pb, Sb, and Zn [2]. In a preceding study at the same site [3], severe heavy metal contamination was identified in groundwater monitoring wells surrounding a waste deposit facility at the Rönnskär Cu–Pb–Zn smelter in Northern Sweden. Utilizing multi-elemental screening

analyses combined with multivariate statistical techniques, it was possible to elucidate the most probable sources and transport pathways of groundwater contamination at this complicated site [3]. The screening analyses revealed the presence of trace elements such as Re and Tl that were not previously known to occur at elevated concentrations at the site. This provided a complete characterization of the different types of waste, as well as the differences among waste deposits and groundwater wells, though it was concluded that in order to obtain more detailed information on specific mechanisms controlling element mobility and source identification in future monitoring programs should include isotope data [3].

Over the last few decades, isotope ratio analysis has become an important tool within the Environmental Forensics' toolbox. For example, tracing anthropogenic sources of lead was accomplished in numerous studies using radiogenic Pb isotopes in a variety of matrices, including mine tailings, discharges from ore processing facilities and urban waste incinerators, flue gases, urban aerosols, sediments, soils, lichens, tree bark, leaves, and clinical samples [4–8]. Improved analytical capability offered by the advent and continued developments of MC-ICP-MS (multi-collector inductively coupled plasma mass spectrometry) instrumentation and analyte purification techniques have introduced non-traditional isotopes, such as Cd, Cu, and Zn, into the field of Environmental Forensics [9–12]. The potential use of these stable isotopes as tracers has been increasingly explored since the identification of isotopic fractionation caused by high-temperature industrial processes [11,13–16]. As isotopic fractionation may occur in many industrial processes with incomplete mass transfer, isotope systems with different degrees of fractionation can be used as powerful tracing tools [17]. Such tracer studies using isotope ratio measurements are increasingly adopting multi-elemental approaches to fingerprint pollution sources [18–20]. However, the applicability of such multi-elemental methods may be limited by the uncertainties of interpreting the isotope ratio data [21].

In the previous studies [6,22] we have assessed the range of isotopic compositions for nine elements in a variety of environmental samples from spatially-limited areas concluding that the natural variability, which often far exceeds the intermediate precision of the analytical methods, needs to be considered when interpreting results of environmental studies. The potential variability in isotopic signatures of pollution sources and post-depositional alteration of isotope ratios can further complicate the interpretation of isotopic information.

Therefore, the aim of this study was to build on the previous work [3] and to extend the investigation by adding environmental samples (brackish water and harbor sediments) from the surroundings of the industrial smelter, as contaminants reaching aquatic environments are predominantly adsorbed to particulate matter and will ultimately accumulate in the sediments close to smelters [7,23–27]. In addition, the ranges of isotopic compositions of four elements (Cd, Cu, Pb, and Zn) in ores, final products, wastes, polluted groundwaters, and sediments were assessed in order to evaluate source variability and possible isotope fractionation at the complex site. If 'isotopic signatures/fingerprints' of the smelter are isotopically distinct from natural background sources (previously evaluated [6,22]), this should aid in deciphering sources and fate of environmental exposure by adding degrees of freedom to the process.

The elements measured in this study were selected as they have been identified as major environmental concerns in the area surrounding the smelter [3]. The elements represent both radiogenic (Pb) and stable isotopic systems (Cd, Cu, and Zn), and can be analyzed using a single multi-element/multi-isotopic analytical procedure [6]. They are all chalcophile elements, naturally present and ubiquitous in soils and sediments. They can be found as major constituents in the ore minerals sphalerite ($ZnS$), chalcopyrite ($CuFeS_2$), and galena ($PbS$). Cadmium is the constituent element of the mineral greenockite ($CdS$) and occurs as a significant impurity (up to 13 wt%) in sphalerite in most Zn-Pb deposits [28–30]. These minerals are commonly mined and refined together and considered the major contamination sources of these elements (other sources being waste incineration, fuel

combustion, chemical industries, and agriculture). Cadmium also has a close relationship with Zn in nature due to their geochemical properties [31]. Cd, Cu, Pb, and Zn's toxicity is well studied, and adverse health effects of human exposure include e.g., brain, liver, and kidney damage [32–35].

## 2. Methods

### 2.1. Study Area

The Bothnian Bay region is sparsely populated but heavily industrialized, with major industries along the Swedish coast including paper mills, a steel foundry (SSAB, Luleå), and the Rönnskär sulfide ore smelter (Figure 1). This study was facilitated by Boliden mineral AB, supplying samples of industrial materials as well as environmental samples collected in the vicinity of the Cu-Pb-Zn smelter at Rönnskär. The smelter is located on the Bothnian Bay coast, 15 km SE of Skellefteå (N 64°40′, E 21°16′), on a largely artificial peninsula, where the two islands Hamnskär and Rönnskär have been joined together using copper slag, including highly permeable Iron Sand (a granulated slag product from the zinc-fuming plant). A more detailed description of the site's geology and hydrogeology can be found in the preceding study [3].

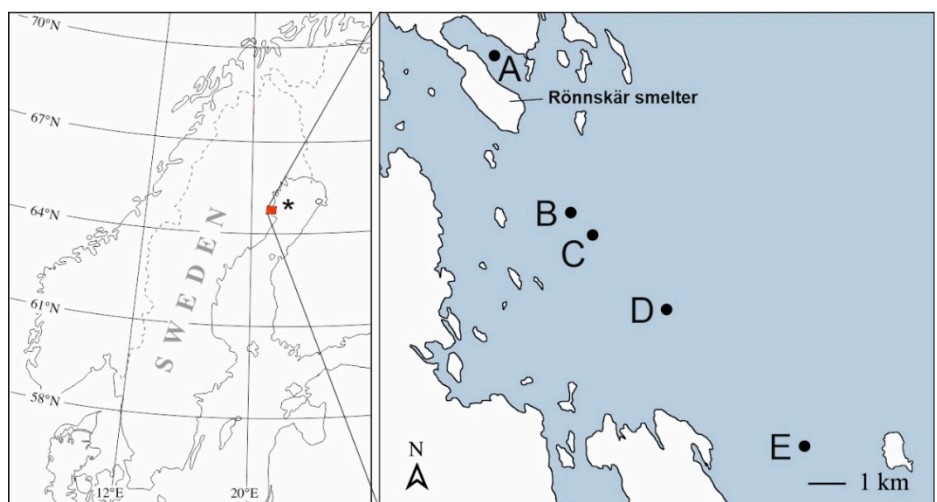

**Figure 1.** Map of the Rönnskär smelter—Bothnian Bay area in northern Sweden. Sampling locations of transect sediment cores are marked A–E. Brackish water sampled in the Skelleftehamn harbor basin (Site A), and Bothnian Bay "clean" reference water at the site marked *.

### 2.2. Production and Waste Generation

The Rönnskär smelter started production in 1930 to processes the complex Cu-As-Au ore from the Skellefte sulfide ore district [36]. A 145 m tall smokestack was constructed at the smelter to disperse airborne pollutants produced from the refining process, and the wastewater generated was discharged into the Bothnian Bay until 1970 when water treatment was introduced. Zinc and scrap metal processing was added to the production line in the 1960s, and in the 1980s, the Rönnskär smelter began to process electronic scrap as demand for Au and Cu increased. Since 2012 the capacity for recycled electronic scrap has surpassed 120,000 tons per year, and ~40% of Au and ~20% of Cu produced at Rönnskär is from recycled scrap materials [37,38]. Today the smelter mainly produces Ag, Au, Cu, Pb, and Zn with $H_2SO_4$ and $SO_2$ as byproducts, as well as granulated slag (Iron Sand) and intermediates (for example, Se and $NiSO_4$). The ore used predominantly originates from the Skellefteå Sulfide Ore District, and the Swedish Aitik and Garpenberg mines, all of which are owned and operated by the Swedish mining company Boliden Mineral AB, which also operates the Rönnskär smelter. However, ores purchased from international mines are also occasionally used [39].

During the 90 years of smelter operation, ore treatment and refining, and the economic situation, have affected both the number of metals produced and the severity of pollution discharge. Due to the site's long industrial history, the area around Rönnskär is particularly well suited for Environmental Forensic studies on base metal pollution and trace element studies. However, its long history also makes Rönnskär an increasingly complex polluting source as feeding materials and production have changed over time [38,40].

Process waste, such as metal-rich dust (produced from smelter operations), and sludge from the water treatment plant, are recycled back through the smelter to further extract precious metals. However, a part of the waste cannot currently be recycled due to the high concentration of contaminants such as, Cd, and Hg. Thus, this waste is stored at designated deposit areas at the Rönnskär site.

Dust produced during gas purification from the copper works and the fumigation plant (hereafter referred to as K1 and F1 dust, respectively) is deposited on the artificial peninsula's western side. The dust was originally mixed with water to create a slurry in order to reduce dust dispersion. However, in 2014 Boliden Mineral AB started storing dry dust in 1000-litter plastic bags. The bags are stored in temporary deposit cells, on a bottom asphalt layer covered with a butyl rubber mat connected to a deposit well, collecting any leachates from the cell. The bags are covered by iron sand, a butyl rubber mat, green liquor dredge, soil, and grass to prevent infiltration of meteoric water and oxidation [41]. Temporary deposition of waste in this area began in 2011 and will continue until the waste is permanently deposited in an underground repository that is currently under construction. The deposits are surrounded by groundwater observation wells.

### 2.3. Samples

Samples of sulfide ore concentrate consisting of chalcopyrite ($CuFeS_2$) from the Aitik Cu mine and complex-ore concentrates containing Au, Cu, Pb, and Zn from the Boliden and Garpenberg ore concentrators were obtained from the Rönnskär smelter. In addition, mono-mineralic (galena, sphalerite, and chalcopyrite) samples from various Swedish ore deposits were obtained from the mineralogical collection of Luleå University of Technology (number of samples, n > 50). To estimate the isotopic composition of elements produced by smelters, a wide range of pure metals (Cu, Pb, and Zn, n = 41), single-element calibration solutions (for all elements, n = 49) and salts (for Cd, n = 4) (hereafter referred to as pure chemicals) were obtained from various suppliers/producers. Time series of K1 and F1 dust samples were collected from gas purification filters at Rönnskär during March 2014 and December 2016 (n = 4). The two dust types were mixed with water to produce a so-called 'pelletization slurry', which is the form in which dust was deposited in the waste storage facility. Leachate was separated from the slurry by centrifugation, which was also sampled (n = 2). A more detailed description of the material flow at the Rönnskär smelter and where the dust is created can be found in the Electronic Supplementary Information (ESI, Figure S1).

Three dust deposit drainage wells were sampled when access was possible (n = 10). Groundwater within the industrial area was sampled from groundwater monitoring wells equipped with polyethylene liners and located at the waste deposit site (n = 76). All wells and deposits were located adjacent to the Skelleftehamn harbor basin with a direct connection to the Bothnian Bay (Figure 1, Site A). Sampling was carried out monthly from May 2016 until December 2016. An intensive second sampling campaign was carried out weekly between 19 May to 16 June 2017.

Grab samples of brackish surface waters were sampled (n = 15) close to the deposit site in the harbor basin (Figure 1, Site A) between the 24th of January to the 16th of June 2017. Samples of unpolluted brackish surface waters (n = 6) were taken from the boat near site E (Figure 1). All water samples were filtered in the field immediately after sample collection into acid-cleaned polypropylene bottles using disposable 20 mL BRAUN Omnifix syringes fitted with SARSTEDT non-pyrogenic sterile filters (pore size 0.45 μm) and stored at 4°C prior to handling and analysis. Five sediment cores (20 cm deep, 8 cm diameter) were

sampled (using a Gemini gravity corer) along a >12 km long NW-SE transect from the Rönnskär smelter into the Bothnian Bay in 2013 (Figure 1, Sites A–E). Each core was divided into four vertical segments, thus representing a record of the recent pollution history.

### 2.4. Sample Preparation

All handling of samples and digests was performed in clean laboratory areas (Class 10,000) by personnel wearing clean-room gear and following all general precautions to reduce contamination risks [42]. Mixed acids digestion was performed on ores, dusts, and sediments (details on the digestion procedure are given in the ESI).

A matrix separation procedure using AG MP-1M ion exchange resin was adopted with only minor modifications from the previous method [6] to provide purified fractions of Cu, Cd, Pb, and Zn for subsequent MC-ICP-MS isotope ratio measurements. In order to reach the required degree of matrix separation, in some samples, the Cu fraction was run through the same column twice. To ensure that no introduction of artificial fractionation occurred, all measurements of samples with recoveries <90% were excluded. Further details on matrix separation are provided in the ESI.

### 2.5. Analytical Methods

A single-collector, an inductively coupled plasma double-focusing sector-field mass spectrometer (ICP-SFMS, ELEMENT XR, Thermo Fisher Scientific, Bremen, Germany), was used to measure element concentrations in water samples, sample digests, and purified fractions after column separation using a combination of internal standardization and external calibration [43–46]. The ICP-SFMS accuracy was evaluated by comparing it with published data for the GBW 07312, Nod-A1, and Nod-P1 CRMs (certified reference materials), presented in Table S1 in the ESI.

A Neptune PLUS Thermo Fisher Scientific, Bremen, Germany MC-ICP-MS instrument was used for the isotope ratio measurements. Instrument operating conditions and measuring parameters are presented in Table 1 and in the ESI. The isotope ratio data's accuracy was evaluated by comparing it with published isotope data for the Nod-A1 and Nod-P1 CRMs, presented in Table S2 in the ESI.

**Table 1.** MC-ICP-MS (multi-collector inductively coupled plasma mass spectrometry) operating parameters and measurement conditions for isotope ratio measurements.

| Elements [a] | Configuration of Introduction System | Resolution Mode | Integration Time (s) | Sample Uptake Rate (L min⁻¹) | Cup Configuration | | | | | | | | |
|---|---|---|---|---|---|---|---|---|---|---|---|---|---|
| | | | | | L4 | L3 | L2 | L1 | C | H1 | H2 | H3 | H4 |
| Cd/Ag | Aridus/Apex desolvating systems, self-aspirating microconcentric PFA nebulizer, X-type skimmer cone | Low | 0.524 | 0.04–0.06 | $^{107}$Ag | $^{108}$Cd ($^{108}$Pd) | $^{109}$Ag | $^{110}$Cd ($^{110}$Pd) | $^{111}$Cd | $^{112}$Cd ($^{112}$Sn) | $^{114}$Cd ($^{114}$Sn) | $^{116}$Cd ($^{116}$Sn) | $^{117}$Sn |
| Zn/Cu | Pumped Micromist nebulizer, double spray chamber, H-type skimmer cone | Medium | 0.262 | 0.20–0.25 | - | $^{63}$Cu | $^{64}$Zn ($^{64}$Ni) | $^{65}$Cu | $^{66}$Zn | $^{67}$Zn | $^{68}$Zn | $^{70}$Zn | - |
| Pb/Tl | Aridus/Apex desolvating systems, self-aspirating microconcentric PFA nebulizer, X-type skimmer cone | Low | 0.524 | 0.04–0.06 | - | $^{202}$Hg | $^{203}$Tl | $^{204}$Pb ($^{204}$Hg) | $^{205}$Tl | $^{206}$Pb | $^{207}$Pb | $^{208}$Pb | - |

[a] One element is used as the internal standard for the second element, except for Ni and Ag, which were only used as the internal standards. RF power: 1400–1450 W. Coolant gas flow: 15 L min⁻¹. Auxiliary gas flow: 1.4 L min⁻¹. Sample gas flow: 0.9–1.25 L min⁻¹. Additional gas flow (N$_2$, Aridus, and Apex) 0.01–0.02 L min⁻¹. Ion lens settings: adjusted daily to obtain maximum sensitivity and signal stability. Zoom optic settings: adjusted daily to obtain maximum resolution. Number of blocks: 9. Number of cycles per block: 5. Number of integrations: 3–5. Amplifier rotation: left. The following chemicals were used as δ-zero standards: NIST SRM 3108 Cd solution Lot 130116, NIST SRM 976 Cu standard solution, and IRMM 3702 Zn solution from the Institute of Reference Materials and Measurements, Geel, Belgium.

## 3. Results and Discussion

Part of concentration data and isotopic information for local topsoils and lysimetric waters was reported in previous studies [3,22]. Discussion in this section examines multi-element patterns and cross-correlations, which help identify major processes affecting all the samples based on previous findings [3]. The multi-element patterns are followed by a section focused on element-specific isotope signatures comparing environmental and industrial samples.

### 3.1. Element Concentrations

#### 3.1.1. Brackish Water

Mean concentrations of selected elements in unpolluted surface water from the Bothnian Bay and brackish waters from the harbor basin are shown in Table 2. A more than tenfold increase in mean concentrations was found in harbor water for As, Cd, Re, and Zn. These elements were also present at high levels in groundwaters from the waste deposition site. However, concentrations of trace elements in the harbor water are several orders of magnitude lower than in the groundwater surrounding the waste deposit area. This difference in concentration is due to dilution in the open Bothnian Bay. There are, however, clusters of elements with high correlations in both matrices with $R^2 > 0.5$. These elements include Ag, As, Bi, Cd, Co, Cu, Fe, Mn, Mo, Ni, Pb, Re, W, and Zn. Since samples were filtered in the field before preservation, Fe and Mn's presence in the cluster suggests that anoxic conditions prevail in the groundwater. Furthermore, after entering surface water, the residence time is expected to be noticeably short for elements that can co-precipitate with Fe and Mn hydroxide particles [47–49]. Large temporal variations observed in the concentrations of trace elements in nearshore brackish waters of the Bothnian Bay during the study period are most likely caused by the implementation of the groundwater pumping wells as a contamination remediation strategy implemented by Vatten och Miljö byrån in 2012 (Vatten & Miljöbyrån, unpublished data).

**Table 2.** Element concentrations in leachate, deposit well waters, groundwaters, Bothnian Bay background reference, brackish water sampled outside deposit facilities, and sediments from the transect outside the Rönnskär smelter. Values displayed as mean *(median)* min–max.

| Element | Leachate | Deposit well Water | Groundwater | Bothnian Bay Reference Water | Brackish Water | Sediments |
|---|---|---|---|---|---|---|
| | *n* = 2 | *n* = 10 | *n* = 40 | *n* = 6 | *n* = 15 | *n* = 20 |
| | mg/L | mg/L | mg/L | µg/L | µg/L | mg/kg |
| As | 5.67 | 454 | 0.21 | 0.05 | 1.7 | 655 |
| | (-) | *(0.2)* | *(0.015)* | *(0.05)* | *(1.7)* | *(244)* |
| | 4.98–7.05 | 0.06–2610 | 0.0003–1.0 | 0.4–0.6 | 1.3–2.6 | 7.5–2850 |
| Cd | 9460 | 3020 | 73 | 0.02 | 2.2 | 21.5 |
| | (-) | *(41)* | *(0.28)* | *(0.02)* | *(0.2)* | *(1.0)* |
| | 9220–9700 | 0.1–19,600 | <0.005–560 | 0.01–0.02 | 0.1–24 | 0.1–314 |
| Cu | 276 | 202 | 0.96 | 0.9 | 2.7 | 458 |
| | (-) | *(112)* | *(8.4)* | *(0.9)* | *(2.5)* | *(128)* |
| | 269–282 | 0.2–799 | <0.0003–7.9 | 0.7–1.1 | 1.9–4.9 | 7.7–2180 |
| Mo | 0.28 | 0.02 | 0.18 | 1.0 | 1.3 | 6.9 |
| | (-) | *(0.03)* | *(0.23)* | *(1.0)* | *(1.1)* | *(3.4)* |
| | 0.27–0.28 | <0.01–0.09 | <0.0004–1.1 | 0.9–1.0 | 1.0–3.6 | 0.5–40 |
| Pb | 80.7 | 26.9 | 0.05 | 0.1 | 0.4 | 366 |
| | (-) | *(2.7)* | *(<0.005)* | *(0.1)* | *(0.3)* | *(102)* |
| | 79.6–81.8 | 4–1730 | <0.0005–0.4) | 0.1–0.2 | 0.1–1.2 | 4–1730 |
| Re | 5.37 | 0.006 | 0.12 | 0.01 | 0.2 | 0.006 |
| | (-) | *(0.001)* | *(0.22)* | *(0.01)* | *(0.3)* | *(0.001)* |
| | 5.36–5.38 | <0.001–0.04 | <0.0005–0.68 | 0.01–0.01 | 0.1–0.3 | <0.001–9.2 |
| Zn | 92,000 | 984 | 880 | 3.5 | 66 | 980 |
| | (-) | *(171)* | *(10.1)* | *(3.2)* | *(12.1)* | *(171)* |
| | 91,600–92,400 | 26.3–9950 | 0.04–6500 | 2.4–5.3 | 8.8–640 | 26–9950 |

### 3.1.2. Sediments

The elevated concentrations observed for many elements in the brackish water near the smelter (Table 2) are expected to affect the sediment composition outside the site. The concentrations for selected elements in sediments are presented as mean, minimum, and maximum in Table 2. The concentrations for all elements measured are presented in Table S3 of the ESI.

The degree of pollution for a given element can be evaluated in different ways. Firstly, it is possible to compare the mean concentration of an element for the core closest to the smelter (site A, Figure 1) with a mean concentration for the more distant core (site E, Figure 2). The highest near-source increase was found for Cd (150 times than that of the more distant core), following by Re (55), Zn (37), Ag (27), and Cu (23). The increase in the range of 10–20 times that of the furthest core was found for As, Hg, Pb, S, and Sn; 5–10 times for Au, Mo, Pd, Pt, Sb, and Te; 2–5 times for B, Ba, Be, Bi, Co, Cr, Ni, Se, and W. For the rest of the elements studied, the concentrations in sediments remain stable irrespective of the distance from the source. This approach's apparent limitation stems from the risk that pollution from the smelter can affect the element concentrations in sediments (top layers especially), even at remote site E.

Secondly, minimum and maximum concentrations found in all sediment samples can be compared, thus reducing the risk of using contaminated sediment layers for normalization and providing information on the extent of the peak pollution. The highest range was again found for Cd (the ratio of maximum to minimum concentration of 4300), following by Te (610), Ag (500), Bi (460), Pb (400), As and Zn (380), Cu (280), Sn (120), and Re (110). The ratios of maximum to minimum concentrations range from 10 to 100 were found for Au, B, Ba, Co, Cu, Hg, Mo, Pd, Pt, S, Sb, Se, and W.

The third approach is based on comparing element concentrations of contaminated sediments to the local background concentrations. The background concentrations are represented by the mean concentrations in the 1.0–5.8 m section of the 5.8 m deep sediment core from the open Bothnian Bay (Figure 1). This core is dated back to 5500 years BP [50]. An enrichment factor (EF) ([Analyte]$_{sample}$/[Analyte]$_{deep\ core\ average}$), is used to estimate the anthropogenic contribution to the total concentration in sediments outside of the Rönnskär site. The mean concentrations of the deep core were as follows; As 12 mg·kg$^{-1}$, Cd 0.16 mg·kg$^{-1}$, Cu 33 mg·kg$^{-1}$, Mo 3.3 mg·kg$^{-1}$, Pb 14 mg·kg$^{-1}$, Re 0.9 µg·kg$^{-1}$, and Zn 107 mg·kg$^{-1}$. A visualization of EF patterns (or element-specific distributions) as a function of depth and distance from the Rönnskär smelter for As, Cd, Cu, Mo, Pb, Re, and Zn is shown in Figure 2. Unsurprisingly, sediments closer to the Rönnskär smelter are significantly more enriched in the elements believed to be associated with its operations. The maximum enrichment for As, Cd, Cu, Pb, Re, and Zn found approximately 15 cm below the sediment surface reflects the effect of the implementation of the recent measures in pollution control.

A rather unexpected finding from the comparison of spatial distributions of several elements has to be mentioned. Rhenium is present in low concentrations in natural environments and behaves conservatively in seawater. In the sediments, Re was found enriched in the core from Site A (Figure 1) with an EF of 14–47, while sediments collected further out in the transect had relatively stable concentrations close to the background. By contrast, Mo, an element originating from the same ore concentrate as Re (Aitik Cu ore), was transported further out into the Bothnian Bay (Figure 2).

The fact that Zn and Cd originate from the same source and share a similar geochemical behavior is reflected in the very close correlation between their concentrations. This similarity occurs in the sediments and all other samples analyzed in this study (Figure 3). Slight variations in the Zn/Cd ratio between sample types could be caused by differences in mobility due to partitioning effects and the formation of secondary minerals, which has been shown in several previous studies [51,52]. Such differences in mobility are expected to be more pronounced for elements with different geochemical behavior. As a result, their

ratios will vary between matrices used for pollution monitoring, which may complicate the use of element ratios as tracers.

Significant ($R^2 > 0.5$) correlations in all sediment samples exist between concentrations of Ag, As, Au, Ba, Cd, Co, Cu, Pb, Pd, Pt, Re, and Zn, forming a cluster with almost identical elements as in groundwater and brackish water. In the most polluted core from site A, two distinct groups of elements can be separated based on significant correlations of their concentrations through sediment column: one consisting from As, Ba, Bi, Hg, Sb, and Se and another from Ag, Au, Cd, Ni, Pb, Pd, Pt, Tl, and Zn. This indicates that the elemental signature of the source varied in the past.

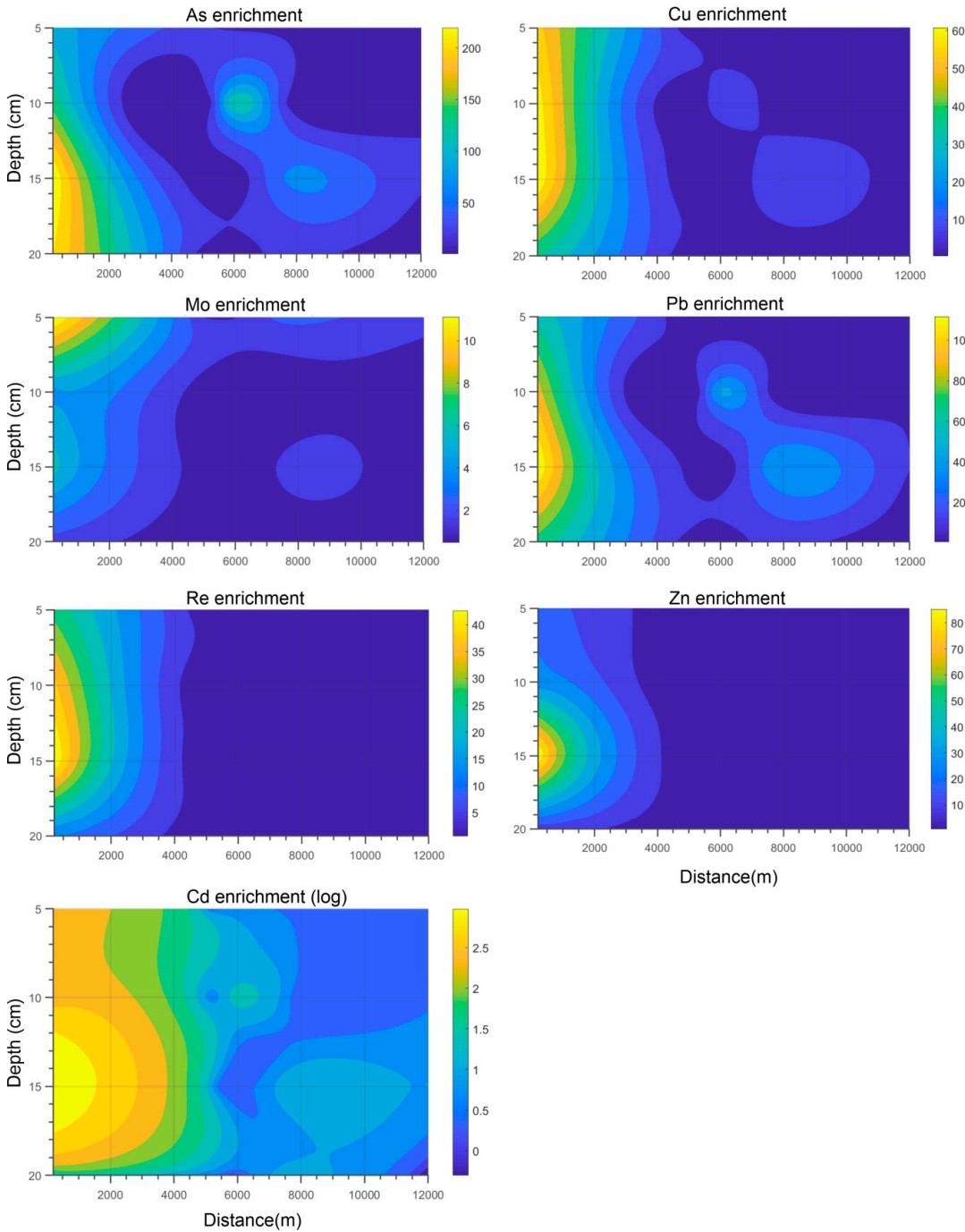

**Figure 2.** Enrichment factors for As, Cd, Cu, Mo, Pb, Re, and Zn using non-linear interpolation between the transect core samples (Figure 1). The enrichment factor (EF) is calculated as $[\text{Analyte}]_{\text{sample}}/[\text{Analyte}]_{\text{deep core average}}$, where "deep core average" is the average concentration in the section between 1.0–5.8 m in a 5.8 m deep core from the open Bothnian Bay [50].

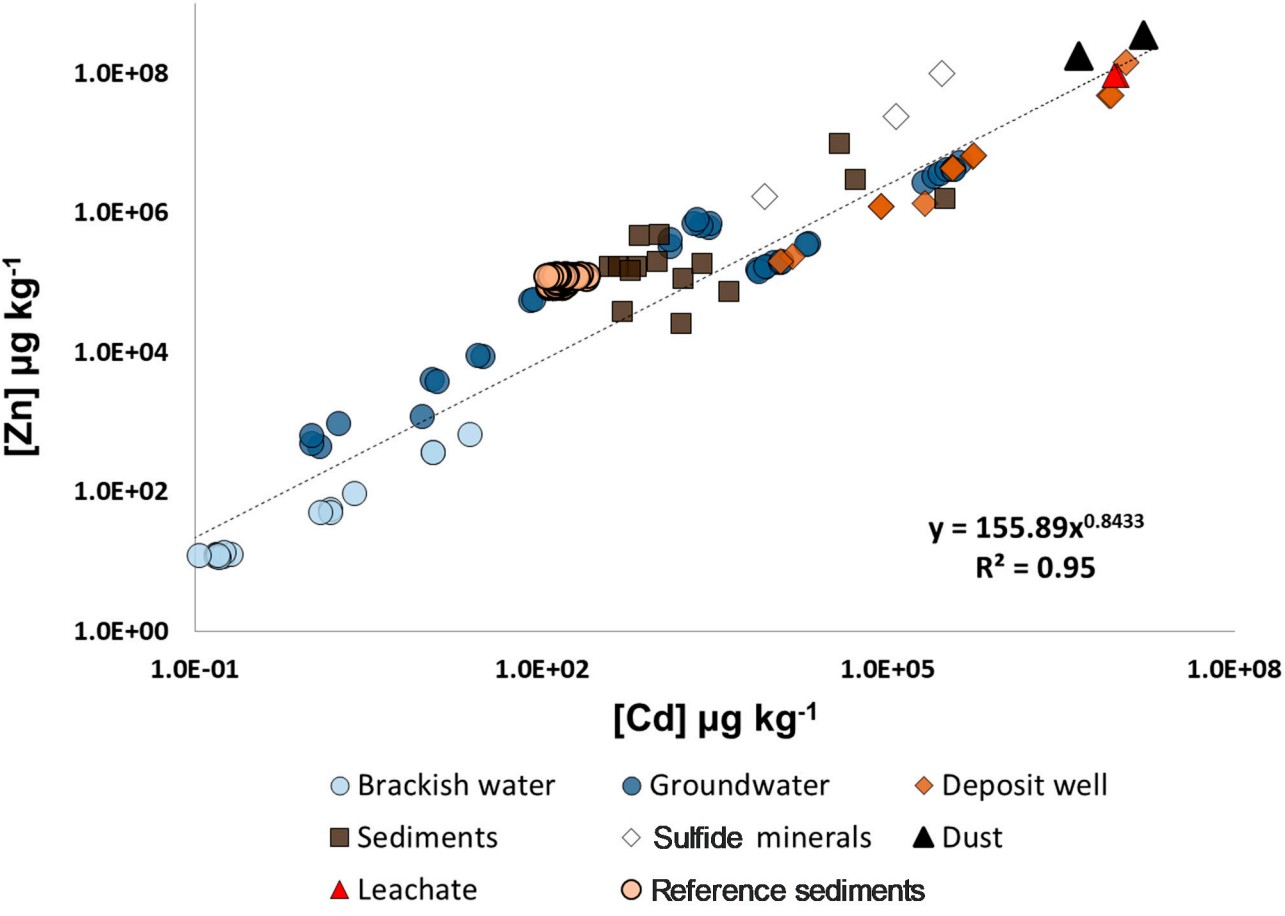

**Figure 3.** Correlation between Zn and Cd in ore concentrates, dusts, leachate, deposit well water, groundwater, and brackish water. Reference sediments from the Bothnian Bay (n > 65, the 1.0–5.8 m section of a 5.8 m deep sediment core from the open Bothnian Bay [50]).

### 3.2. Isotope Ratios

Summary of isotopic information for Pb, Zn, Cd, and Cu is presented in Tables 3–6 for topsoils, lysimetric waters, sulfide minerals, pure chemicals, K1 and F1 dust samples, leachates, deposit well water, groundwater, and sediments. Topsoils and lysimetric waters were included as they are considered a major source of near shore sediments from relatively unpolluted areas, and therefore may estimate local background isotopic compositions for studied elements. The existence of differences in mean isotopic compositions of stable elements between these two matrices would indicate fractionation during the soil leaching process—a factor that must be considered when evaluating the leaching of dust samples. Isotopic composition of sulfide minerals and pure chemicals characterize potential isotopic changes in material flows entering and leaving a smelter. Sulfide minerals can also be considered as a possible direct pollution source for near-shore sediments by losses during transport through and unloading in the harbor. Deposited dust was shown to be the major pollution source of these four elements in groundwaters from the area surrounding the smelter [3]. Isotopic characterization of leachate, deposit well water, and groundwaters provides an opportunity to study potential post-deposition isotopic fractionation during dust leaching, migration of contaminated waters, and secondary mineral formation. Finally, sediments serve as a reservoir for various inputs from the area where isotopic signatures of individual sources are blended. A very simplified mixing situation with two isotopically distinct sources—natural (with a low concentration of elements studied) and anthropogenic

(with high concentrations)—should result in a linear regression line on a graph where the isotopic composition is plotted as a function of inverse element concentration. The intercept of the *y*-axis would then provide the isotopic composition of the anthropogenic source. One obvious missing link between different potential pollution sources and sediments is brackish water with no isotopic information available. Regretfully, a combination of limited volumes of brackish water sampled, relatively low analyte concentrations, and the challenging matrix have prevented reliable isotope ratio measurements from being performed in this matrix.

**Table 3.** Lead isotopic composition in environmental, geological, and industrial samples. Matrices marked * reported from previous studies [22].

| | | $^{206}Pb/^{207}Pb/^{208}Pb/^{207}Pb$ | | | |
|---|---|---|---|---|---|
| **Matrix** | **Number** | **Mean (SD)** | **Min.** | **Max.** | **Range** |
| Topsoils * | 150 | 1.35(0.15)/2.49(0.10) | 1.05/2.30 | 1.77/2.61 | 0.72/0.31 |
| Lysimetric waters * | 15 | 1.17(0.03)/2.45(0.02) | 1.10/2.42 | 1.20/2.47 | 0.10/0.05 |
| Sulphide minerals | 47 | 1.05(0.10)/2.33(0.08) | 1.00/2.29 | 1.52/2.71 | 0.52/0.42 |
| Pure chemicals | 27 | 1.10(0.07)/2.36(0.07) | 1.03/2.29 | 1.18/2.46 | 0.15/0.17 |
| Dust K1 | 4 | 1.11(0.01)/2.38(0.02) | 1.10/2.37 | 1.12/2.39 | 0.02/0.02 |
| Dust F1 | 4 | 1.10(0.02)/2.37(0.02) | 1.08/2.35 | 1.11/2.39 | 0.03/0.04 |
| Leachate | 2 | 1.10(0.01)/2.37(0.01) | 1.10/2.37 | 1.11/2.38 | 0.01/0.01 |
| Sediments | 20 | 1.18(0.06)/2.41(0.03) | 1.08/2.36 | 1.31/2.48 | 0.23/0.12 |

**Table 4.** Zinc isotopic composition in environmental, geological, and industrial samples. Matrices marked * reported from previous studies [22].

| **Matrix** | **Number** | $\delta^{66}Zn$, ‰ **Mean (SD)** | **Min.** | **Max.** | **Range** |
|---|---|---|---|---|---|
| Topsoils * | 150 | 0.18(0.20) | −0.47 | 0.69 | 1.18 |
| Lysimetric waters * | 15 | 0.15(0.13) | −0.05 | 0.30 | 0.35 |
| Sulphide minerals | 56 | 0.08(0.12) | −0.10 | 0.25 | 0.35 |
| Pure chemicals | 35 | 0.13(0.22) | −0.25 | 0.57 | 0.82 |
| Dust K1 | 2 | −0.61(0.06) | −0.65 | −0.57 | 0.08 |
| Dust F1 | 3 | −0.67(0.10) | −0.73 | −0.56 | 0.17 |
| Leachate | 3 | −1.06(0.05) | −1.11 | −1.00 | 0.11 |
| Deposit well water | 9 | −1.21(0.20) | −1.53 | −0.95 | 0.58 |
| Ground water | 6 | −1.40(0.16) | −1.56 | −1.18 | 0.38 |
| Sediments | 20 | −0.24(0.24) | −0.85 | 0.02 | 0.87 |

**Table 5.** Cadmium isotopic composition in environmental, geological, and industrial samples. Matrices marked * reported from previous studies [22].

| **Matrix** | **Number** | $\delta^{114}Cd$, ‰ **Mean (SD)** | **Min.** | **Max.** | **Range** |
|---|---|---|---|---|---|
| Topsoils * | 150 | 0.08(0.14) | −0.32 | 0.42 | 0.74 |
| Lysimetric waters * | 15 | 0.22(0.07) | 0.14 | 0.34 | 0.20 |
| Sulphide minerals | 38 | −0.06(0.12) | −0.17 | 0.08 | 0.25 |
| Pure chemicals | 16 | −0.05(0.15) | −0.58 | 0.02 | 0.60 |
| Dust K1 | 2 | −0.12(0.01) | −0.12 | −0.11 | 0.01 |
| Dust F1 | 3 | −0.13(0.05) | −0.18 | −0.11 | 0.07 |
| Leachate | 3 | 0.15(0.02) | 0.14 | 0.17 | 0.03 |
| Deposit well water | 6 | 0.12(0.05) | 0.06 | 0.21 | 0.15 |
| Ground water | 5 | 0.00(0.10) | −0.13 | 0.09 | 0.20 |
| Sediments | 20 | −0.16(0.14) | −0.48 | −0.04 | 0.44 |

**Table 6.** Copper isotopic composition in environmental, geological, and industrial samples. Matrices marked * reported from previous studies [22]. Marked NA when Not Applicable.

| Matrix | Number | $\delta^{65}$Cu, ‰ Mean (SD) | Min. | Max. | Range |
|---|---|---|---|---|---|
| Topsoils * | 150 | 0.06(0.21) | −0.51 | 0.51 | 1.02 |
| Lysimetric waters * | 15 | 0.76(0.17) | 0.64 | 0.99 | 0.35 |
| Sulphide minerals | 47 | 0.40(0.54) | −0.72 | 1.60 | 2.32 |
| Pure chemicals | 18 | 0.38(0.78) | −0.46 | 1.58 | 2.04 |
| Dust K1 | 2 | 0.26(0.06) | 0.21 | 0.30 | 0.08 |
| Dust F1 | 2 | 0.04(0.03) | 0.02 | 0.06 | 0.04 |
| Leachate | 2 | 0.13(0.03) | 0.11 | 0.15 | 0.04 |
| Deposit well water | 3 | 0.98(1.23) | −0.10 | 2.07 | 2.17 |
| Ground water | 1 | 1.66 | NA | NA | NA |
| Sediments | 19 | −0.31(0.26) | −0.94 | 0.20 | 1.14 |

### 3.2.1. Lead

Lead is the only radiogenic isotope system included in the present study. As mass-dependent fractionation of Pb isotopes is negligible compared to the radiogenic signatures and the fact that Pb ores (and therefore industrial Pb) often have distinct isotopic signatures that are different from the natural background, Pb isotope ratios were used for source identification for several decades [53–57].

Extreme variability in Pb isotope composition seen in topsoils (with a range of $^{206}$Pb/$^{207}$Pb ratios corresponding to almost 50% of the mean value, Table 3) reflects Pb sources with quite different isotope signatures. As has been shown previously [22], soils from relatively unpolluted areas have high $^{206}$Pb/$^{207}$Pb ratios (in excess of 1.3). This is consistent with the isotopic composition of Pb in the local bedrock [57], while polluted soils tend to have lower $^{206}$Pb/$^{207}$Pb ratios. This is caused by aeolian sources (assessed from an analysis of lichens), which have a $^{206}$Pb/$^{207}$Pb ratio of 1.14. This ratio is close to the mean ratio of lysimetric waters, suggesting relatively high mobility of Pb atmospheric components in soil.

Despite relatively wide ranges of Pb isotopic compositions found in the Swedish sulfide minerals, most minerals with high Pb concentrations (thus making an important contribution to the smelter's element inflow) have very non-radiogenic signatures resulting in mean $^{206}$Pb/$^{207}$Pb of 1.05. However, mean $^{206}$Pb/$^{207}$Pb ratio in pure chemicals, dusts, and leachate is slightly higher (1.10–1.11), most probably reflecting the contribution of lead from recycling electronic scrap.

Lead isotope ratios in sediments display a clear ($R^2 = 0.47$) correlation with inverse Pb concentrations (Figure 4) consistent with mixing between a 'natural' Pb with more radiogenic ratios and an anthropogenic one with less radiogenic ratios. The mixing can be further visualized in a three-isotope plot ($^{208}$Pb/$^{207}$Pb vs $^{206}$Pb/$^{207}$Pb, Figure 5) in which Pb isotopic ratios plot in between mean value for topsoils and sulfide minerals. However, among sediment samples with the highest element concentrations, only few have isotopic signature identical to one found for dust samples (Figure 4). The intercept of the regression line (1.16) can probably serve as an accurate assessment for mean $^{206}$Pb/$^{207}$Pb ratio of the anthropogenic lead emitted by the smelter over recent operation history.

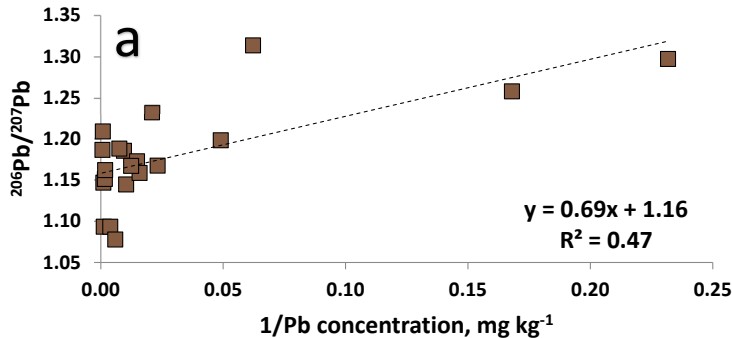

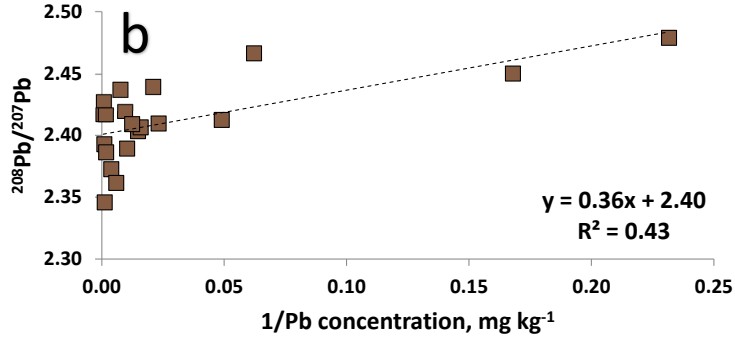

**Figure 4.** (**a**) Correlation between radiogenic $^{206}$Pb/$^{207}$Pb signatures and 1/Pb concentration in sediments collected along the 12 km transect. (**b**) Correlation between radiogenic $^{208}$Pb/$^{207}$Pb signatures and inverse Pb concentration in the same sediments. Measurement error is smaller than the size of the data point markers.

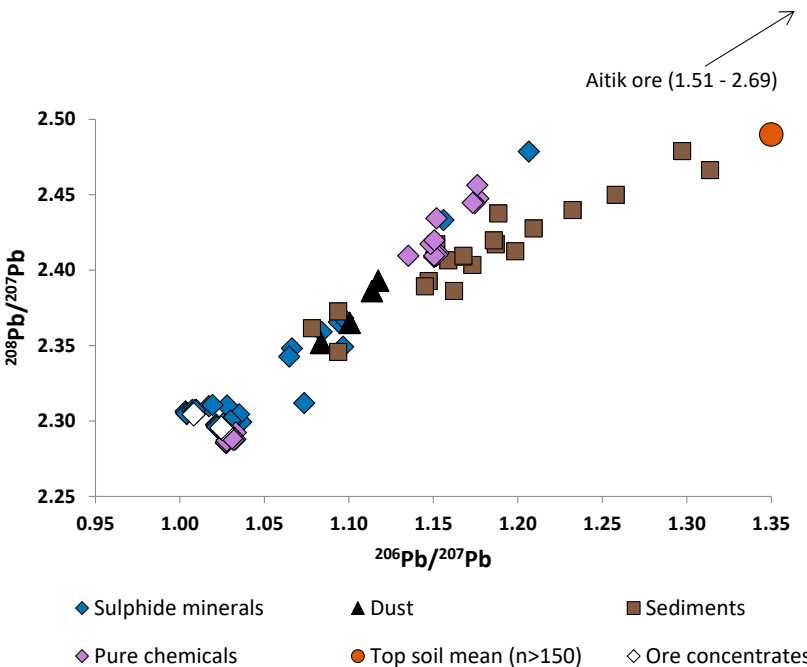

**Figure 5.** Three isotope plots ($^{208}$Pb/$^{207}$Pb vs $^{206}$Pb/$^{207}$Pb) of Swedish sulfide mineral collection, pure chemicals, ores, dusts, topsoils, and sediments. One ore sample, Aitik ore is positioned outside of the figure ($^{208}$Pb/$^{207}$Pb = 1.51; $^{206}$Pb/$^{207}$Pb = 2.69) indicated but the arrow. The topsoil marker represents the mean for n > 150 samples, previously reported [22]. Measurement error is smaller than the size of the data point markers.

### 3.2.2. Zinc

In contrast to the wide range of Pb isotope ratios in ores, the $\delta^{66}$Zn of minerals and ore concentrates tested within a relatively narrow range ($-0.10$ to $0.25‰$, with a mean of $0.08‰$). These limited variations demonstrate a uniform isotopic composition of the Rönnskär smelter feeding materials. The mean $\delta^{66}$Zn of topsoils and lysimetric waters is approximately $0.2‰$, which is very similar to those in the ores concentrates (Table 4). However, the published ranges of $\delta^{66}$Zn for sulfide minerals of major ore deposits worldwide are much broader ($-0.96$ to $1.06‰$) [58–62].

Excluding one Zn calibration solution having $\delta^{66}$Zn of almost $11‰$, the mean $\delta^{66}$Zn for pure chemicals ($0.13‰$) is almost identical to that of the ore concentrates (Table 4). This is likely a reflection of the high Zn recovery (yield) during modern metallurgical processing, limiting the possibility for significant fractionation between the source and refined metal. Published isotopic compositions of common metallic Zn products also fall into relatively narrow ranges ($-0.17$ to $0.03‰$ [63]), confirming insignificant isotopic fractionation between ores and the final metallic products.

The isotopic composition of Zn in the K1 and F1 dusts is significantly lighter than that of the ore concentrates and Zn chemicals (Table 4), which is expected given that the former is affected by evaporation known to favor light isotopes [15,27,64]. A similar Zn isotopic composition was reported in samples collected from the chimneystack ($-1.00$ to $-0.90‰$) of a French smelter [64].

Interestingly, $\delta^{66}$Zn of the leachate (representing the mobile water-soluble fraction of Zn in K1 and F1 dust) is even more negative ($-1.06‰$) (Table 4). This suggests an additional fractionation that discriminates heavy isotopes during the Zn dissolution process, despite no significant differences being apparent between the mean isotopic composition of topsoils and lysimetric waters (Table 4). A further shift towards lighter Zn isotopic compositions occurs in deposit drainage wells and then in groundwaters (mean $\delta^{66}$Zn of $-1.21‰$ and $-1.40‰$, respectively, Table 4). The observed fractionation is likely caused by a combination of several processes, such as preferential adsorption of heavy isotopes on inorganic surfaces (e.g., hematite [65], as well as higher mobility of light isotopes during diffusion [66]. Regardless of the exact nature and relative contribution of underlying processes, a gradual enrichment of light Zn isotopes is seen. This enrichment occurs during the ore concentrates–dust–leachate transition and then during water migration through isolation barriers and soils. The experimental data clearly depicts the isotopic signature(s) of anthropogenic Zn released by the smelter being very different from that of the natural background. However, post-deposition fractionation results in rather broad ranges of $\delta^{66}$Zn (e.g., almost $0.6‰$ for well water), rather than a well-defined "isotopic fingerprint".

The most heavily polluted harbor bay sediments at approximately 15 cm depth in the vicinity of the smelter (Figure 2) have the lightest $\delta^{66}$Zn at $-0.85‰$, and a strong positive correlation ($R^2 = 0.82$) between the inverse Zn concentration and $\delta^{66}$Zn (Figure 6a). The intercept of the regression line in Figure 6a points towards an anthropogenic source with $\delta^{66}$Zn of approximately $-0.94‰$. This is between the mean values for dusts (as a result of airborne contamination) and contaminated groundwater (Table 4). It is possible that an additional shift in the Zn isotopic composition may be induced from adsorption processes of dissolved Zn on the surfaces of settling particles. When including Zn data for all five sediments cores, the correlation became much less significant ($R^2 = 0.17$, Figure 6b). It seems that the isotopic composition of the least polluted sediments approaches that for soils and lysimetric waters. However, there are many sediment layers with elevated Zn concentrations and with $\delta^{66}$Zn undistinguishable from that for sulfide minerals, suggesting the possibility of pollution from the handling of ore concentrates in the area.

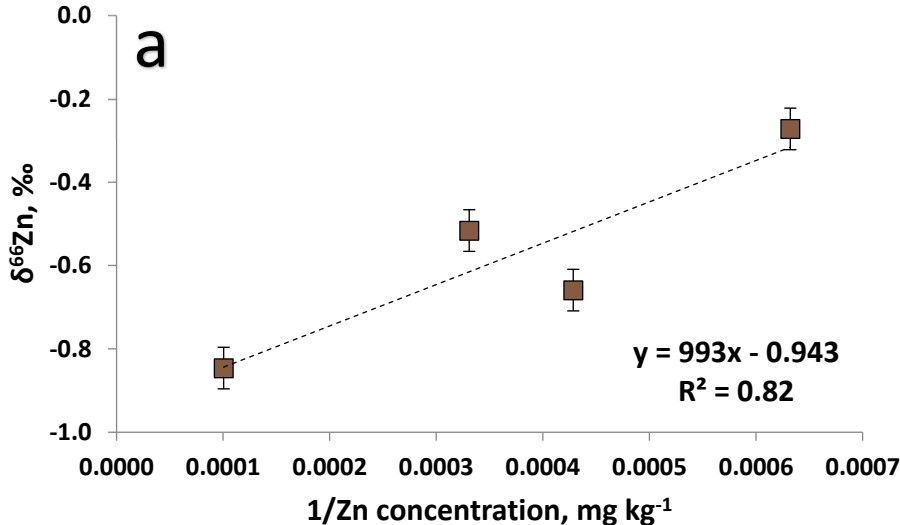

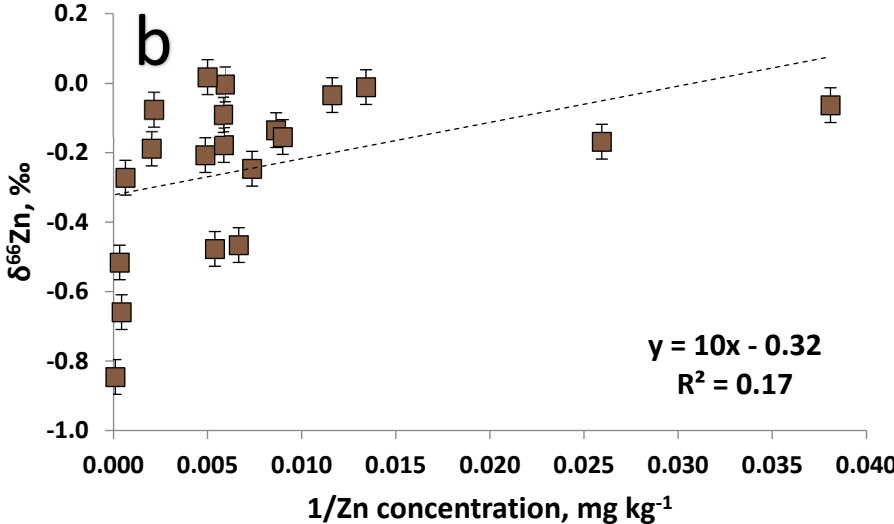

**Figure 6.** (**a**) Correlation ($R^2$ = 0.82) between $\delta^{66}$Zn signature and 1/ Zn concentration in a sediment core collected in the harbor bay close to the Rönnskär smelter (Site A, Figure 1). (**b**) The correlation is significantly weaker ($R^2$ = 0.17) when including all sediment samples from the 12 km transect.

### 3.2.3. Cadmium

As in the case of Zn, there are very limited Cd isotopic variations in minerals/ore concentrates processed by the smelter ($\delta^{114}$Cd ranges from −0.17‰ to 0.08‰, with a mean of −0.06‰, Table 5). The range of the isotopic composition in topsoils is slightly broader ($\delta^{114}$Cd ranges from −0.32‰ to 0.43‰, with a mean of 0.08‰). This is in excellent agreement with published ranges for $\delta^{114}$Cd in terrestrial rock and soil samples varying from −0.4‰ to 0.4‰ [13,67]. Mean $\delta^{114}$Cd values for pure chemicals and sulfide minerals are almost identical, suggesting little or no fractionation during the manufacturing. However, both types of dust produced at the Rönnskär smelter are enriched in light Cd isotopes compared to the feeding materials (Table 5). Published ranges of industrial wastes are relatively wide ($\delta^{114}$Cd varying from −0.64‰ to 0.46‰, with slags on the heavier side and aerosols and dusts produced in high-temperature processes on the lighter side [14,30,68].

In contrast to Zn isotopes' leaching behavior, leaching of dust with water preferentially releases heavier Cd isotopes with an isotopic difference between the solid and liquid phase of approximately 0.3‰. It should be noted that the mean Cd isotopic composition of lysimetric water is also 0.14‰ heavier than that of bulk soils (Table 5). Preferential partitioning

of heavy Cd isotopes into a liquid phase was previously reported in experiments involving sorption to Mn-oxides [69], calcite precipitation [70], and phosphate fertilizer application experiments [71]. The drainage wells collecting excess water from the Rönnskär waste deposits have almost identical Cd isotopic composition as the leachates, while groundwaters have a slightly lighter Cd isotopic composition. This probably due to a fractionation effect from diffusion that theoretically should benefit lighter isotopes as was the case for Zn.

The majority of the sediment samples have a relatively uniform Cd isotopic composition spanning from the lighter $\delta^{114}$Cd values measured in dust samples to the heavier values found in sulfide minerals (Figure 7). However, a few samples with elevated Cd concentrations have $\delta^{114}$Cd values as low as −0.48‰. The experimental data on fractionation during sorption of Cd to Mn-oxide surfaces, with lighter isotopes preferentially sorbed and heavier isotopes remaining in solution [69], may contribute to the isotopically light Cd observed in the polluted sediments closest to the smelter.

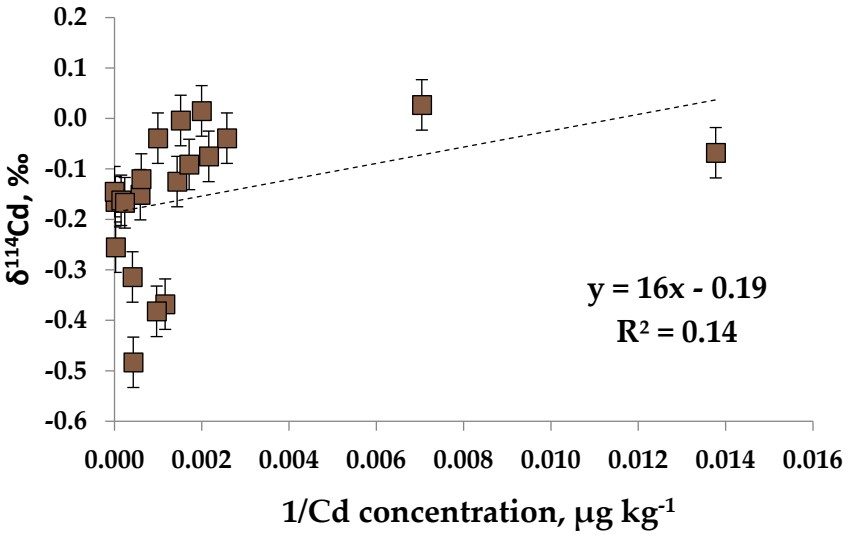

**Figure 7.** Correlation between $\delta^{114}$Cd signatures and 1/Cd concentration in the sediment core collected along the transect outside the Rönnskär smelter.

### 3.2.4. Copper

The mean $\delta^{65}$Cu for topsoils in the area of 0.06‰ (Table 6) agrees well with the isotopic composition reported for terrestrial igneous rocks [9,72–74]. A significantly heavier Cu isotopic composition found in lysimetric waters ($\delta^{65}$Cu = 0.76‰) reaffirms previously published findings showing that the heavy isotope of Cu is preferentially partitioned into aqueous phases [10].

Compared to $\delta^{66}$Zn and $\delta^{114}$Cd, considerably wider ranges of $\delta^{65}$Cu were found in both sulfide minerals (from −0.7‰ to 1.6‰) and pure chemicals (from −0.5‰ to 1.6‰) (Table 6). These ranges are, however, narrow compared with the published ranges for ores and hydrothermal systems (from −4‰ to +9‰, [75–79]). Despite this isotopic variability, the mean $\delta^{65}$Cu values for minerals and pure chemicals are almost identical (Table 6).

The isotopic compositions of Cu in K1 and F1 dust samples are both well within the source minerals range, corroborating findings on low degrees of Cu isotopic fractionation in smelting processes [64]. In contrast to significant differences in the Cu isotopic composition between topsoils and lysimetric water, no such differences exist between K1 and F1 dusts and leachate (Table 6). However, already in deposit well waters mean $\delta^{65}$Cu reaches almost 1‰, with further fractionation in favor of heavier Cu in the groundwater (1.7‰). This shift towards higher $\delta^{65}$Cu values in the order leachate—deposit well waters—groundwaters are accompanied by a significant decline in dissolved Cu concentrations [3]. This suggests the existence of mechanisms preferentially removing isotopically lighter Cu from the system.

The Cu isotopic composition of sediments does not reflect the predominantly positive $\delta^{65}$Cu values found in ores/concentrates, dust, and, especially, polluted waters at the waste deposit site. The majority of sediment samples have $\delta^{65}$Cu in a narrow range from $-0.2$ to $-0.4$‰ (Figure 8, Table 6), while two samples from core A in the vicinity of the smelter have identical values of $-0.94$‰. Negative $\delta^{65}$Cu signatures have previously been reported in soil systems affected by anthropogenic Cu [16].

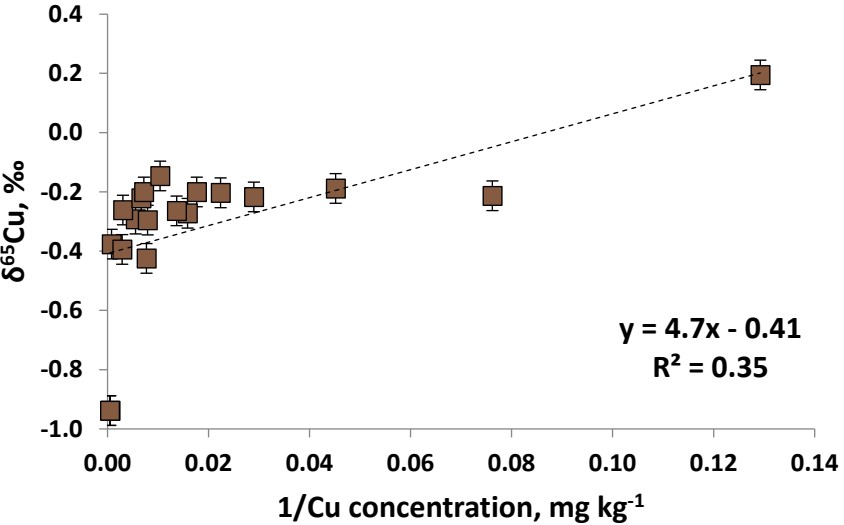

**Figure 8.** Correlation between $\delta^{65}$Cu signatures and 1/Cu concentration in the sediment core collected along the transect outside the Rönnskär smelter.

## 4. Conclusions

Similar clusters of elements with elevated concentrations and significant inter-element correlations can be traced practically in all matrices tested (from dust samples to sediments), confirming a link between the source and the polluted environmental compartments. Differences in the relative mobility in the environment for different elements (shown in the example of Re and Mo distribution in sediments) may affect the usefulness of the elemental ratios in reconstructing the extent and timing of pollution events.

The Pb in the sulfide minerals used by the smelter and the elevated element concentrations in the sediments has an isotope composition that is significantly different from the local geological background, reaffirming the potential of this radiogenic system in Environmental Forensics investigations. However, large temporal variability of the smelter feeding material may affect Pb isotopic signatures over time and hampers the effectiveness of traditional fingerprinting.

The industrial processes result in wastes (dust) with isotopically light Zn, and the light isotope signature is further amplified during post-depositional processes, which can be traced all the way into the sediments. Regardless of the exact nature of underlying fractionation processes, the signature(s) of anthropogenic Zn released by the smelter is distinguishable from natural sources.

The degree of Cd isotopic fractionation in dust caused by the high-temperature process is significantly less pronounced than that of Zn. Both the environmental and industrial leaching favors heavy Cd isotopes, but the following transport of dissolved Cd causes a gradual shift to a lighter isotopic composition, most likely due to fractionation during diffusion. This lighter Cd signature can then be found in polluted sediments, but with a narrower isotopic span than that for Zn.

The wide variability in the Cu isotopic composition found in the smelter feeding materials reflects the large ranges observed in ores in Sweden and worldwide. However, in the dust waste and the leachate, the ranges are narrow. The Cu isotopes are significantly heavier in the deposit wells and groundwater, most likely as a result of preferential

incorporation of light Cu isotopes into solid phases as the isotopic shift is accompanied by a significant decline in dissolved Cu concentrations. The light Cu isotope signatures observed in the polluted sediments closest to the smelter suggest further changes in the isotope composition occurring between groundwater and deposition in sediments.

Among elements evaluated in this study, radiogenic Pb and stable Zn isotope systems offer the most promising source identification in the area close to the smelter. However, temporal variability in the isotopic composition of the source adds complexity for the former. Numerous post-deposition fractionating processes alter the original source ratios for Cu, Zn, and to a lesser extent, Cd. At larger distances from the source, additional fractionation during element migration and dilution of source-specific signatures with background components makes source tracing more challenging.

For the great promise offered by expanding the number of elements utilized in isotope tracing as a powerful way to decipher sources and fate of environmental exposure to be fully realized, a comprehensive evaluation of both source(s) and background variability, as well as post-depositional fractionation, needs to be an integral part of any Environmental Forensics investigation.

Finally, there are still several areas that need improvement and future work:

1. Isotopic information in electronic scrap and brackish waters is missing, making the source's characterization and evaluating potential fractionation during transport between groundwater and sediments incomplete.
2. The sampling of dust, leachates, groundwater, and brackish water should be performed over longer periods and more frequently to better assess temporal concentrations and isotopic compositions.
3. By increasing the length and horizontal resolution of sediment cores, it is possible to provide a more accurate and detailed record of local pollution history.
4. The addition of Fe, S, Cr, and Tl isotope systems may help better understand post-depositional isotopic fractionation.

Those may, to some degree, affect the certainty of the conclusions and are needed to be taken into consideration during the planning of future studies.

**Supplementary Materials:** The following are available online at https://www.mdpi.com/2075-163X/11/1/37/s1, Table S1. Inductively coupled plasma double-focusing sector-field mass spectrometer (ICP-SFMS) results for reference materials. Where not stated otherwise, concentrations are in mg kg$^{-1}$ for solids and in µg L$^{-1}$ for waters; Table S2. Isotope composition of Cd, Cu, Pb, and Zn in Nod-A1 and Nod-P1 CRM; Table S3. Element concentrations in sediment samples, mg kg$^{-1}$, instrumental precision: 3–7% RSD; Figure S1. A simplified process flow sheet demonstrates where the F1-dust and the K1-dust are obtained in the respective production line.

**Author Contributions:** Conceptualization, S.P. and E.E.; Formal analysis, C.P. and I.R.; Funding acquisition, E.E. and A.W.; Investigation, S.P. and A.M.; Methodology, C.P.; Project administration, E.E., A.W., A.M. and I.R.; Resources, S.S.-J., K.R. and C.P.; Writing—original draft, S.P.; Writing— review & editing, S.P., S.S.-J., K.R. and I.R. All authors have read and agreed to the published version of the manuscript.

**Funding:** This research was funded by The Swedish Agency for Economic and Regional Growth, grant number 20202314. The APC was funded by Luleå University of Technology.

**Acknowledgments:** This study is part of the Waterface project, funded by the Swedish Agency for Economic and Regional Growth and the County Administrative Board of Norrbotten, Sweden. We wish to thank the two companies participating in the project, ALS Scandinavia AB and ÅF Infrastructure, for invaluable technical support. We also thank Pasi Peltola, Environmental Coordinator at the Rönnskär smelter, for providing samples and background information on the study site, as well as valuable input during the project. The authors also wish to express their gratitude to the environmental sampling crew at Rönnskär, who made the intensive groundwater sampling program possible, and to Susanne Bauer and Sarah Conrad for providing the reference water samples from the Bothnian Bay and Luleå University of Technology for providing access to the geological collection.

Nicola Pallavicini and Anna Stenberg are acknowledged for the optimization of the separation method used in the study.

**Conflicts of Interest:** The authors declare no conflict of interest. The funders had no role in the design of the study; in the collection, analyses, or interpretation of data; in the writing of the manuscript, or in the decision to publish the results.

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
