# Peer review of "Evaluation of a Multi-Isotope Approach as a Complement to Concentration Data within Environmental Forensics"

_minerals, doi:10.3390/min11010037_

Round 1

Reviewer 1 Report

Comments and Suggestions for Authors:

The paper titled “Evaluation of a multi-isotope approach as a compliment to concentration data within Environmental Forensics” deals with heavy metal contamination in groundwater monitoring wells surrounding a waste deposit facility at the Rönnskär Cu–Pb–Zn smelter in Skellefteå, Northern Sweden, as well as in brackish water and sediments from the nearby harbor. The aim of the study was to evaluate the variability of the polluting source and identify possible isotope fractionation.

In general, the manuscript is well structured and comprehensive. The subject of the research is of great interest. What the authors want to say in the present study is well said and stated. The research is properly organized and conducted. After all, in the authors there are included some of the most significant researchers concerning this subject. Hence, their contribution is apparent throughout the whole manuscript.

In the “Introduction” part, the authors describe the reasons why this research was conducted. The discussion and the interpretation of the results are appropriate and detailed enough. In the supplementary material all the needed information concerning details of the followed methodology is included. Nevertheless, it would be auspicable if the authors would provide more clearly the objectives of the present study in their introduction. Regarding the English language style, although I do not feel qualified to judge about it, it must be said that it is very comprehensive and minor spell check is required. For these reasons, minor revision should be carried out before publication, while some comments and suggestions for the authors are listed below. It would be preferable if the authors would take them into account.

“Introduction” part:

Only one reference is referred at the end of each one of the first 3 paragraphs. It would be auspicable if the authors could increase the number of the cited references at this point of such an interesting and informative introduction.

Row 41:

“In a preceding study at the same site,”: the corresponding study should be better cited here than at the end of the whole sentence.

“Study area” part:

Since the manuscript is concerning the concentrations of specific metals in samples of groundwater, brackish water and sediments, it is obvious that the geological setting of the study area is important. Hence, maybe the authors could consider providing information regarding it. Moreover, they should consider the possibility of also including a geological map of the study area, apart from the geographical map in Figure 1.

Rows 203-208:

In these sentences mean concentrations, elements present at high levels and lower concentrations of trace elements in the harbor water than in the groundwater are mentioned. It would be better if apart from the description, the authors could also mention the corresponding values. That kind of information would make the discussion more interesting to the reader. The same should be considered throughout the whole manuscript, as well.

Figure 4:

Please pay more attention to the diagrams. Some of their parts are overlapped.

As a general comment, maybe the authors could consider the possibility of enhancing the tables in the manuscript and make them more comprehensible to the reader. In some cases they are a little bit confusing, especially since in some cases where more than one rows in a column correspond to one row of other columns. Just make them more clear. In addition, maybe the authors should consider the possibility of the graphical presentation of the results. There is a lot of numerical information and too much data obtained by the research, but they are mainly referred in tables. If the results were demonstrated in diagrams, the manuscript would become much more comprehensive to the reader.

Author Response

We thank the reviewer for the thoughtful comments and suggestions for improvements, here follows a point by point list of changes and answers to suggestions made.

“Introduction” – We have added more references to the mentioned 3 paragraphs as suggested

”Row 41” – Citing has been improved, preceding study is now cited where it was first mentioned.

“Study area” – We agree with the reviewer, the geology is of importance. The geology has been described in the preceding study, hence we added a reference to where it can be found so readers who are interested can obtain it, as well as the hydrogeology.

”Rows 203-208” – We understand the reviewers comment, however with the large amounts of matrices and elements analyzed, we feel that mentioning individual values makes the text significantly harder to digest for readers. The stylistic choice of condensing information into tables were therefore made.

“Figure 4” – Yes, something went wrong with the template formatting here, hopefully this will get sorted before the final version.

“Final note” – Again similar to the comment on “row 203-208”, we understand the reviewers thoughts. We made multiple attempts on the structuring of such a large dataset on a wide variety of sample types and concentrations/isotopes. We found that the stylistic choice of condensing the information into tables made the text most enjoyable for readers and simultaneously kept a stronger theme through the article, while also giving anyone the opportunity to further use our results in future.

Reviewer 2 Report

Section 1. Introduction

Very good presentation of the reasons as well as aim of study, including complementarity with other published papers. Congrats to the team for the extended work and impressive number of data collected during the study.  

Section 2. Methods

2.3. Samples

- while explaining how samples were collected the letter n is used, most probably to show the number of samples that were collected; I suggest simply mention the significance of n at its first use

- locations and time period of sampling are mentioned; however, considering the results interpretation and correlations made in further sections, some clarity needs to be added with regards to the sampling moment, where possible – frequency of the sampling, if from same location all sample categories (as described in Table 2) were collected in the same day or week, etc

Section 3. Results and discussion

  • row 203 – mean concentrations were mentioned as included in Table 2, however I did not find a mention saying if each result reported comes from a duplicate / triplicate measurement; Table 3 also mention mean values but I found no info on number of samples measured; suggestion - to be mentioned;
  • rows 214-217 – large temporal variations are mentioned; however Table 2 does not give information on timing, and the text in section 2 needs more clarity in this regards

ESI

  • row 21 – precision of weighind needs to be included, as well as material of 50 mL tubes mentioned
  • row 23 – concentrations of HF to be mentioned, as well as type of heat block used
  • row 24 – that 1 mL HNO3 added was of what concentration
  • ro2 25 – clarity on technical parameters needs to be added (i.e. type of agitation / equipment, centrifugation – speed, etc
  • table 3 – data on measurement precision need to be included

Author Response

We thank the reviewer for the thoughtful comments and suggestions for improvements, here follows a point by point list of changes and answers to suggestions made.

“Samples”

1 – The use of “n” for numbers of samples has been clarified where its used for the first time, as suggested.

2 – We agree with the reviewer mentioning the lack of timing and frequency of sampling where long term sampling were made, proper information regarding our sampling campaing has been added as suggested under the "2.3 Samples" section.

“Section 3. Results and discussion”

“Row 203” – We think there was some confusion here to what “mean concentration” refers to. The mean concentration reported and discussed is mean values from samples measured, for example n=40 groundwater samples, the mean between all 40 samples collected is reported as a mean value in the table.

However, each of those 40 groundwater samples are indeed measured in triplicates. The mean value the reviewer mentions (duplicates/triplicates) is related to the precision of the method and instrumentation.

Regarding the instrumentation operating conditions and parameters, and more detailed subjects such as precision and matrix effects are described in great detail in the references used in the analytical methods section (mainly ref 44, 45, 46) as well as the ESI. This information has knowingly been removed from the main article as they have been published numerous times, and due to the large scope of the paper the analytical method section has been made shorter. 

“Table 3”, "mention mean values but I found no info on number of samples measured” - Again there seem to be some confusion similiar to earlier question on row 203, as the number of samples are reported in column 2 for each matrix analyzed.

“Row 214-217” – We agree with the reviewer mentioning the lack of timing and frequency of sampling where long term sampling were made, proper information regarding our sampling campaing has been added as suggested under the "2.3 Samples" section.

“ESI”

Row 21-22 - Precision of the scales, and tube material added as suggested

Row 23 - Concentration of HNO3 added as suggested

Row 24-25 - Agitation device and centrifugation speed added as suggested

Table 3 - Instrument precision added as suggested